# Majorana Anyon Composites in Magneto-Photoluminescence Spectra of Natural Quantum Hall Puddles

**DOI:** 10.3390/nano12061016

**Published:** 2022-03-20

**Authors:** Alexander M. Mintairov, Dmitrii V. Lebedev, Alexey S. Vlasov, Steven A. Blundell

**Affiliations:** 1Ioffe Institute, 194021 Saint Petersburg, Russia; lebedev.dmitri@mail.ioffe.ru (D.V.L.); vlasov@scell.ioffe.ru (A.S.V.); 2Electrical Engineering, University of Notre Dame, Notre Dame, IN 46556, USA; 3SyMMES, IRIG, CNRS, CEA, University Grenoble Alpes, F-38000 Grenoble, France; steven.blundell@cea.fr

**Keywords:** quantum dots, magneto-photoluminescence, fractional quantum Hall effect, anyons, Majorana modes, topological quantum computing

## Abstract

In magneto-photoluminescence (magneto-PL) spectra of quasi two-dimensional islands (quantum dots) having seven electrons and Wigner–Seitz radius *r_s_*~1.5, we revealed a suppression of magnetic field (*B*) dispersion, paramagnetic shifts, and jumps of the energy of the emission components for filling factors *ν* > 1 (*B* < 10 T). Additionally, we observed *B*-hysteresis of the jumps and a dependence of all these anomalous features on *r_s_*. Using a theoretical description of the magneto-PL spectra and an analysis of the electronic structure of these dots based on the single-particle Fock–Darwin spectrum and many-particle configuration-interaction calculations, we show that these observations can be described by the *r_s_*-dependent formation of the anyon (magneto-electron) composites (ACs) involving single-particle states having non-zero angular momentum and that the anyon states observed involve Majorana modes (MMs), including zero-*B* modes having an equal number of vortexes and anti-vortexes, which can be considered as Majorana anyons. We show that the paramagnetic shift corresponds to a destruction of the equilibrium self-formed *ν*~5/2 AC by the external magnetic field and that the jumps and their hysteresis can be described in terms of Majorana qubit states controlled by *B* and *r_s_*. Our results show a critical role of quantum confinement in the formation of magneto-electrons and implies the liquid-crystal nature of fractional quantum Hall effect states, the Majorana anyon origin of the states having even *ν*, i.e., composite fermions, which provide new opportunities for topological quantum computing.

## 1. Introduction

Superconducting (dissipationless) transport corresponding to zero electrical resistance of the materials, discovered in 1913 [1], gave a quantum mechanical illustration of Newton’s first law of motion, which states that moving material objects will conserve a constant velocity if no external force acts on them, and which represents an obvious paradox for our imagination, since natural movements observed in everyday life are subject to a frictional force and are slowed down. Thus, in superconducting materials, the persistent currents loops (PCLs) can be thought to be perpetual and laboratory measurements show their life-time to be at least 100,000 years [2]. In an external magnetic field (*B_e_*), the persistent vortex currents induce a perfect diamagnetism, i.e., the expulsion of *B_e_* from the superconductor (SC), known as the Meissner effect [3]. PCLs in SCs provide the quantization of the magnetic flux [4] in multiples of *ϕ*_0_^*^ = *h*/2*e*_0_, where *e*_0_ is the electron charge, which was documented in the measurements of the magnetization of micrometer size SC cylinders [5,6]. Single *ϕ*_0_^*^ vortexes, also called Abrikosov vortexes or fluxons, arranged in regular lattices, are generated in major SCs above the critical field *B_e_*_1_ [7,8]. The vortexes consist PCLs tens of nanometers in size, whose density is proportional to *B_e_*, and they merge at a critical field *B_e_*_2_ > *B_e_*_1_, transferring the material into a resistive state. Thus, the existence of *ϕ*_0_^*^ is directly related to the existence of systems having SC-type dissipationless transport. Such transport corresponds to a suppression of the scattering (“friction”) of charged carriers and exists in systems having a gap in the density of conducting states, which in conventional SCs arises from the formation of the bound electron pair state known as a Cooper pair [9]. 

While it is commonly accepted to associate superconductivity with Cooper-type electron pairing, a zero electrical resistance, i.e., SC-type dissipationless transport, is observed in two-dimensional (2D) electron (*e*) gas semiconductor structures in the quantum Hall effect, i.e., at high perpendicular *B_e_*, at integer (IQHE) [10] and fractional (FQHE) [11] Landau level (LL) filling factors *ν = nϕ*_0_/*B_e_*, where *ϕ*_0_ = 2ϕ_0_^*^ and *n* is the electron density. This transport is provided by a skipping-type edge current [12], which for the FQHE involves composite quasi-particles, called composite fermions (CFs) [13], consisting of *e* with 1/*ν ϕ*_0_ magnetic-flux-quanta vortexes (*V*s) attached, resulting in a fractional charge equal to *νe*_0_. The presence of the *V*s and SC-type gapped state, however, implies PCLs generated by CF *e*. While the necessity of such intrinsic sub-*e* PCLs was not considered in the commonly used description of the FQHE states, which is based on Laughlin’s approach [14], their existence was revealed in our recent observation of FQHE-type states for single [15] and five [16] electrons confined in quasi-2D InP/GaInP_2_ islands-quantum dots (QDs) having dimensionless Wigner–Seitz radius *r*_s_~4 and ~2, respectively, where *r*_s_ = 1/[*a*_B_^*^(*πn*)^0.5^] and *a*_B_^*^ is a Bohr radius.

The FQHE-type states have been observed in these QDs at zero *B_e_* using high-spatial-resolution magneto-photoluminescence (magneto-PL) spectroscopy in the measurements of the *B_e_*-dispersion of PL spectral components and their spatial localization. The appearance of these states results in a built-in magnetic field *B_ν_*~6–15 T and a dependence of the ground-state, equilibrium fractional charge *ν*_0_*e*_0_ on the dot size *D*, for which *ν*_0_ decreases with *D*. These observations imply a self-generation of *V*s in a quantum state |i〉 occupied by a single *e*, and a fixed size of the *V* (at zero external field *B_e_*) equal to 2*a_B_*^*^. The *e*, thus, can move without dissipation within a distance *r*_s_*a_B_** and for *r*_s_ > 1 can be considered as nano-superconducting puddle, having charge-density distribution $\rho \left(r\right)={\left|\psi_{i}\left(r\right)\right|}^{2}$, where $\psi_{i}\;$is the wave-function of the state |i〉. Thus, $\psi_{i}\;$can be formally considered to be a complex SC order parameter of the linearized Ginzburg–Landau equation [17] used for the description of mesoscopic SC structures [18,19]. This single-*e-*puddle can support ~*r*_s_ sub-*e* PCLs generating 1/*ν*_0_ *V*s. This reduces the Coulomb interaction energy of *e*s and, thus, the state having 1/*ν*_0_ sub-*e* PCLs and charge *ν*_0_*e*_0_ is self-formed [16]. The *ν*_0_ and corresponding *B_ν_* = *n*2*ϕ*_0_*ν*_0_ are directly determined by the dot size (*D*) and the number of *e*s (*N*), i.e., *r_s_*. We called such a state a magneto-electron (*em*) and denoted it *e^ν^*, where the *em* composition *ν* = *n*/*k*, with *n* (*k*) the number of *e*s (*V*s), is the quantity that substitutes the LL filling factor of the FQHE for quantum confined *e*s. We observed *em*s having *ν* < 1, *n* = 1–4, *k* = 1–9 and molecular structures$\;6e^{1/4}$, $5e^{1/4}$, $3e^{2/7},$ and $3e^{1/3}+e^{4/15}$, having *e^ν^* size ~40 nm and bond length ~60 nm for *N* = 5 and 6. The last two structures are transformed into one another by photoexcitation involving the braiding and fusion of *e^ν^*s, which are the elementary topological quantum computing (TQC) operations [20]. 

We should point out that our observations of the fractional charge and the self-formation of *V*s of a single *e* at zero magnetic field sound like a paradox (similar to the perpetual persistent current in SC) or an artifact, not taking into account the quantum nature of single *e* states, for which non-zero angular momentum states consist of “fractional charge” parts, and relying on the famous experiment of Millikan demonstrating a “fixed” *e*_0_ value in charged oil droplets [21]. However, at the same time, in parallel experiments made by Ehrenhaft, much smaller charges, i.e., sub-*e*s, have been revealed in metal (Au, Pt, …) particles (see description of Millikan-Ehrenhaft dispute on sub-*e* in Ref. [22]). Moreover, much later, a transfer of 1/3*e*_0_ charge from tungsten to Nb balls has been detected in experiments on superconducting magnetic levitation [23]. We can suppose that these observations can be connected to our observation of *em*. 

Nevertheless, since we observed *e^ν^ em*s in QDs having *ν* < 1, one can expect their formation in QDs having *ν* > 1, as for FQHE states [24]. These states correspond to partial (total) filling of the highest (lower) LLs and involve CFs, which are anyons [16], in the highest LL and IQHE *e*s in the lower one. 

A much investigated *ν* > 1 FQHE state is the *ν =* 5/2 state, which, according to the conventional description, corresponds to the half-filled second LL and fully filled lowest one. The CFs in the second LL show non-Abelian anyon properties [25], which implies that they have half-flux *V*s with a zero-energy excited state, known as a Majorana zero mode or simply a Majorana mode (MM) [26,27,28,29], and thus are a perspective for the realization of TQC [30,31]. 

In *em,* the MM is a particle-antiparticle pair, consisting of *V*s having an opposite, anti-parallel direction/orientation. It can be naturally generated in a state having non-zero angular momentum *l_z_*, for which the *e*-wave-function has “anti-phase spatial splitting”. In such states, only half of the $\psi_{i}$ can generate *V* along the *B_e_* direction, which thus is a half-*V* having a MM excited state. For the *em* the MM adds two more *V*s resulting in a two-times smaller charge, the signature of which was observed for the 5/2 state [32]. Moreover, since a MM adds zero magnetic flux, it can possibly be generated at zero *B_e_*. 

A realization of self-formed *ν*_0_ > 1 *em*s/anyons in general and *ν*_0_ = 5/2 in particular requires a decrease of *r*_s_. In our earlier studies [33]*,* we have measured a quantum Hall-type InP/GaInP_2_ QD having *N*~9, *r*_s_~1.5 and *B_ν_*~2 T, which approximately matches that expected for *ν*_0_ = 5/2. 

Here, we analyze a few such dots in detail using measurements of the *B_e_*-dispersion of their PL spectra components. In the measurements, we have revealed several anomalies, which include suppression of the dispersion, paramagnetic shifts, jumps (up to ~2 meV), and their hysteresis. We found that the appearance of these anomalies depends on the dot size and the direction of change of *B_e_*. We analyzed the experimental data using a theoretical description of PL spectra and the electronic structure of such dots based on single-particle Fock–Darwin (FD) states and a many-particle configuration-interaction (CI) approach, which allows one to explain the observed anomalies by the self-formation of *em*-MM-anyon composites corresponding to *ν*_0_~5/2, their collapse and the emergence of the anti-*em*-MM composites induced by the magnetic field. Our analysis has shown that the formation of the *em*s corresponds to the generation of *V*s by single and pair non-interacting *e*s occupying quantum-confined states. This allows one to describe the FQHE in terms of liquid-crystal states involving Majorana anyons and opens new routes for the realization of topological quantum computing.

## 2. Materials and Methods

### 2.1. Natural Quantum Hall Puddles 

The details of the growth procedure and structural and emission properties of the InP/GaInP_2_ QDs were described previously [34]. The dots have a flat lens shape (aspect ratio 10) and a lateral size *D*~50-180 nm. Their shape reveals a small elongation and asymmetry, which in most cases can be described as a combination of a ∼5% elliptical distortion (*D_‖_*/*D*_⊥_ = 1.05) and a 10% change of *R*_⊥_. The specific structural property of this QD system [35] is an atomic ordering of the GaInP_2_ matrix material, which results in composite core-shell structure consisting of InP QD having zinc blende crystal structure surrounded by a few atomically ordered GaInP_2_ domains having rhombohedral crystal structure and size 10-50 nm. In this composite, the domains generate strong piezo-electric fields resulting in *e* doping (up to 20) and *B_ν_* (up to 15 T) forming natural quantum Hall puddles.

Here, we studied three of such puddles denoted D01m, D07m and D09m. They have a number of electrons *N* = 6 and 7 and *D*~75 nm resulted in the electron density *n*~3 × 10^11^ and *r*_s_~1.3, which corresponds to a weak Wigner localization regime. 

### 2.2. Single Dot Magneto-Photoluminescence Spectroscopy Measurements 

We measured magneto-PL spectra of single puddles using a home-made near-field scanning optical microscope (NSOM) having spatial resolution up to 25 nm operating at 10 K and magnetic fields of up to 10 T. We used home-made tapered fiber probes coated with Al, having an aperture size of 50–300 nm in a collection-illumination mode. The spectra were excited by the 514.5 nm Ar-laser line (Edmundoptics, Cherry Hill, NJ, USA) and measured using a CCD (multi-channel) detector (Horiba, Piscataway, NJ, USA) together with a 280 mm focal length monochromator (Horiba, Piscataway, NJ, USA. The excitation power measured before fiber coupler was ~5 *μ*W, which provided a power density of ~0.5 W/cm^2^. The spectral resolution of the system is 0.2–0.4 meV. *σ*^+^ and *σ*^−^ circular polarization were measured using λ/4 plate (Newport, Irvine, CA, USA) and linear polarizer (Newport, Irvine, CA, USA). In the spectra, we measured the shifts of PL peaks related to occupied *e-*shells versus *B_e_*, which were compared with the theoretical calculations and used to extract the charge of single particle states of the puddles.

### 2.3. Analysis of the Data

The shape of the PL spectra, i.e., the number of spectral components, their position and full-width-at-half-maxima, were analyzed using a multi-peak fitting procedure from graphic software OriginPro (version 20.0, Northampton, MA, USA) 

The values of effective (screened) quantum confinement *ħ**ω*_0_^*^ (see Section A.3) and *N* were measured from the splitting and the number of anti-Stokes peaks in the PL spectra, respectively. For *N,* the measurements of the shifts of anti-Stokes components in magnetic field were used as well. The *D* values were estimated from the charge density distributions (CDD) calculated from the unscreened quantum confinement value *ħ**ω*_0_ = *ħ**ω*_0_^*^/0.7 and *N* values using CI approach [16,34] and from NSOM scanning experiments. Using *N* and *D* values we calculate *n* and *r*_s_. Note that these values are related to the photo-excited state having *N** = *N* + 1 electrons. 

### 2.4. Theory and Calculations

The theoretical description of the PL spectra of quantum Hall puddles in a magnetic field based on the Fock–Darwin (FD) spectrum is presented in Appendix A. The description involves a general model (Section A.1), an analysis of the single-particle states of InP/GaInP_2_ QDs involved in PL transitions using the 8 band $\bar{k}\cdot \bar{p}$ method, i.e., non-interacting electrons and holes (Section A.2), and the analysis of the effect of the Coulomb interaction on single-particle states using the Hartree–Fock (HF) method (Section A.3). The resulting formulas of the model, i.e., *B_e_*–dispersion of PL peaks, are Equations (A4c) and (A5) in Section A.4. 

The ground states (GSs) of the puddles in a magnetic field are analyzed in Appendix B. Total energy and total angular momentum values were calculated using the multi-particle CI method (Section B.1) and the single-particle FD spectrum, including the vortex contribution (Section B.2). Both methods revealed GSs having fractional *ν*, suggesting *em*-composite (*em*-C) formation. 

The parameters of *em*-Cs, which are composition *ν_n_*^+^(*ν_n_*)*_S_*, where *n* is number and *S* is a total spin, and charge *e_N_^*^* are specified in Section C.1 (see Equations (A6)–(A8)). There, a mismatch of the emission energy of *em*-Cs is analyzed and corresponding quantities, which are Δ*_ν_* jumps and related PL shift (*E*_s_*^em^*^-C^), are described (Equations (A9) and (A10)). 

The fitting of the Fock–Darwin spectrum for the measurements of *ν* and details of the construction of the *V* structure of *em*-Cs are described in Section C.2 and Section C.3, respectively.

### 2.5. Summary of Measured Parameters

In Table 1, we summarize the initial zero-field parameters of the puddles, which are the number of the electrons in photo-excited state *N** = *N* + 1, *D*, *r_s_*, *ħω*_0_^*^/*ħω*_0_, and the final parameters of *em*-Cs formed in these puddles obtained from the measurements and the analysis of the data, which are two intrinsic magnetic fields *B*_ec_ and *B_ν_* (see below), and *ν*_0_(*ν*_0_^+^)*_S_* and *ν_n_*(*ν_n_*^+^)*_S_* values of the equilibrium and *B_e_*-induced *em*-Cs, respectively. In the notation of the compositions in Table 1, we include reduced charge value *e_N_*^*^ as a superscript.

## 3. Experimental Results

### 3.1. Shell Structure and Built-In Magnetic Field of Quantum Hall Puddles

Figure 1a compares the PL spectra of the dots/puddles studied at zero internal field *B* (see below), plotted in Stokes energy units. The inserts show CI CDDs of the dots and a contour plot of spatially resolved PL spectra near the center of D07m dot. Figure 1b shows circularly polarized spectra of the D07m dot measured at *B_e_* = 0, 1, 2, …, 10 T.

CDDs show a decrease of the size related to a decrease/increase of *D*/*ħ**ω_0_* (see Table 1). The CDDs are elongated along *x* and their landscape has the same topology consisting from two minima of ~*D*/6 size separated along *x* by ~*D*/3*,* which indicates molecular structure (see Section B.1). The left insert shows the measured size of the emission area of the D07m dot of ~90 × 60 nm (average size of 75 nm), in agreement with the corre sponding CDD size.

In Figure 1a, it is seen that the spectra of the dots studied consist of the main peak denoted by *s* and two weaker anti-Stokes ones denoted by *p* and *d*. These three peaks are related to three occupied *e*-shells of the QD in the photo-excited state, as described in Section A.1. The peaks have a full-width-at-half-maximum ~3 meV and varying splitting corresponding to *ħω*_0_^*^ (see Table 1), related to the size variation. The D09m dot also has a few times larger relative intensity of *p*- and *d*-components. There is an increase of the ratio of the *s-p* to *p-d* splitting related to the size increase. 

In Figure 1b, the PL spectra of the D07m dot at zero *B_e_* reveals nearly a two-times stronger intensity of the *σ^−^*- component, i.e., they are *σ^−^*-polarized, which indicates *B_ν_*. At *B_ec_* = 3 T, the *σ^−^*-polarization disappears and the spectra acquire a *σ^+^*-polarization at larger *B_e_*. This indicates *B_ν_~−B*_ec_ (see Table 1), where *B*_ec_ is a compensating field and *B_ν_* has a direction opposite to *B_e_*. Thus, in the range from 0 T to *B*_ec_, the internal field *B = B_e_* + *B_ν_* is negative and decreases from *B_ν_* to zero. In this range, which we denote *B^−^*↓ or *B*_e_^a^, *B_e_* < |*B_ν_*| and the PL spectra have an anomalous *B*_e_-shift (see below). Such an anomalous shift is a direct signature of *B_ν_*, which allows one to detect it independently from polarization measurements. For larger fields, i.e., *B_e_* > |*B_ν_*|, *B* becomes positive and increases versus *B_e_*. In this range, which we denote by *B^+^*↑ or *B*_e_^n^, normal *B*_e_-shifts are observed, but *B* is lower *B_e_* on *B_ec_*. 

There is an anomalous *B*_e_^a^-range, indicating *B_ν_* in D01m and D09m dots too.

### 3.2. Shell Structure in Magnetic Field 

In Figure 2a–c, we present unpolarized spectra of the dots, measured at *B_e_* = 0, 1, 2, … 10 T. The spectra were measured under an increase of the field from 0 to 10 T, denoted as *B_e_*↑ and shown in the lower part of the graphs, and a decrease from 10 to 0 T, denoted as *B_e_*↓ and shown in the upper parts of the graphs. For this range, the *B*_e_^n^ and *B*_e_^a^ ranges are the *B^+^*↓ and *B^−^*↑–ranges, respectively. 

The spectra having *B* = 0 T and dividing the *B_e_*^a^- and *B*_e_^n^-ranges (*B*_ec_ = 3 and 2 T was found in D01m and D09m dots, respectively) are shown by thick solid lines. In Figure 2a–c, the peak maxima are connected by straight lines, which allows one to quali tatively trace their shift. 

At all *B_e_* values, the *s*-peak is dominant, and changes of the spectral shape are caused by the changes of the intensity and the position of the *p*- and *d-*peaks at *B_e_* > 4 T. These are: the peak merging at *B_e_* = 7 T in the D01m dot; the few-times intensity increase of the *d*-peak at *B_e_* = 9 T and the appearance of the additional *x*-peak between the *s*- and *p*-peaks at *B_e_* = 7 T in D07m dot; a few-times intensity decrease of the *p*- and *d*-peak at *B_e_* = 8 T in the D09m dot. 

The peak shift, further denoted as *s*, *p*- and *d*-shifts, in the *B_e_*^n^-range is mostly positive, i.e., increases with field increase, and at *B_e_* = 10 T reaches average values of ~2, ~5, and ~3 meV, respectively. In the *B*_e_^a^-range, the shifts are negative (paramagnetic), i.e., “anomalous”, and have values from 0 to 2 meV. In both ranges, the shifts reveal a few bends (see spectra near *B_e_*~3 T and 6 T). 

### 3.3. Anomalous Shifts and Jumps Induced by Magneto-Electrons 

In Figure 3a–c, we show the measured data points of the peak positions versus *B* and calculated *B*-dispersion of the peaks overlaid on the contour plots of the spectra. Calculated shifts are presented in the *B*_e_^n^-range, i.e., for B ≥ 0 T. The peaks are denoted by the quantum numbers of the Fock–Darwin (FD) spectrum *kl* (see Section A.2). 

The *s*-shift data show that the bends seen in the spectra in Figure 2a–c are jumps having amplitude 1–2 meV over Δ*B_e_*~1 T (see dashed ovals in Figure 2a–c). Their appearance is different in different dots and *B/B_e_* ranges, revealing size dependence and *B_e_*↑–*B_e_*↓ asymmetry (hysteresis). In the D01m dot, one step is observed in both *B^−^*↓– and *B^−^*↑–ranges at *B = −*1 T, which is thus symmetric. In the D07m dot, two asymmetric steps are observed, one in the *B^−^*↓– and the other in the *B^+^*↓–ranges at *B = −*2 and 5 T, respectively, and no mirror steps appear in the ranges of reversed direction of field change. In the D09m dot, four asymmetric steps are observed in the *B^−^*↓– and *B^+^*↑–ranges at *B = −*2, 3, 5 and 7. 

The calculated *s*-shift shows a weak, nearly linear dispersion with slope ~0.25 meV/T and in the regions where the jumps are absent the experimental slope is the same as the calculated one. 

The *p*-shift follows a dispersion of *10*-state up to *B_c_*~5 T and at larger field the shift is saturated (for D01m dot) or becomes negative (for D07m and D09m) approximately reaching the *02–04* states, i.e., shows bending related to level crossing (see Figure A2c). The *p*-shift has a ~30% smaller slope than the *01*-state (~1 meV/T). 

The *d*-shift follows the *11-*state for *B_e_* < *B_c_* for all dots and the *03*-, *04*- and *05*-states for *B* > *B_c_* and the D01m, D07m and D09m dots, respectively, revealing a level crossing similar to the *p*-shift. For the D01m dot, the *d*-shift is strongly suppressed compared to the *11*-state.

In the *B*_e_^a^ range, the experimental data points do not coincide with mirror counterparts in the *B*_e_^n^-range, i.e., the shifts have a *B*_e_^a^*/B*_e_^n^ asymmetry. This is most clearly seen for the *d*-peak in the D01m dot. 

The appearance of the *B_e_^a^-*range is direct evidence for the existence of the *em*s in this range. For the *B_e_^n^-*range, the existence of the *em*s follows from the observation of the jumps of the *s*-peak and the suppression of the *B_e_*-dispersion of the *p*- and *d*-peaks for *B_e_* < *B_c_*. The former are Δ*_ν_* steps and the latter is direct evidence for charge reduction of the *e* in the single-particle state (see Section C.2).

Since the *B* has opposite directions in the *B*_e_^a^ and *B_e_*^n^ ranges, the *V*s and *em*s of the former are anti-*V*s (*aV*s) and anti-*em*s (*aem*s) in the latter (see Section C.1). The *em*s at zero *B_e_* are self-organized and can be considered to be equilibrium *em*s, which we will denote as *S*-*em*s. Thus, the shifts of the peaks in the *B*^a^_e_-range are related to the destruction of *S*-*em*s by the external field, which implies that the internal field generated by *S*-*em*s can be *B_ν_ ≠ B*_ec_ and the *B* decrease versus *B*_e_ increase in this case is non-linear in the *B*_e_^a^-range, as observed. The *em*s in the *B_e_*^n^ range are induced by the *B_e_* and can be considered as stimulated, further denoted as *B*-*em*s.

### 3.4. Majorana Anyon Composites in Fock-Darwin Spectrum

In Figure 4a–c, we present the results of the e-em-FD spectrum fit to the experimental data. The data and fit are presented in the reduced field ν_B_^-1^ and energy ω/ω_0_ units (see Figure A2c) and the e-em-FD-states (e-em-FDSs) are labeled as e^ν^_l_, where l is angular momentum of the state (see Section C.1). In the figures, we also show total charge values e_N_^*^ (upper plots) and constructed V-structures (inserts). Total charge values are shown for photo-exited and initial states and were used to calculate the Δ_ν_ and E_S_^em-C^. The experimental data for B_e_↓ and B_e_↑ measurements are plotted in the same graph and they reveal a B_e_↑–B_e_↓ asymmetry of the s-peak jumps as discussed above, and the same for p-shift in D07m and D09m for ν_B_^−1^~0 and 0.4–0.9, respectively. 

A comparison of the plots in Figure 4a–c shows different ν_B_^−1^ ranges, acquired using the same B_e_ range, in the dots studied, which is due to the different dot size giving an increase of the ν = 1 field B_0_ as D decreases (see Section B.1). The final ν_B_^−1^value is 0.6 for the smallest D01m dot and 1.1 for the largest D09m dot. At the same time, the ν_B_^−1^ starting value is the same ~−0.29, i.e., ν~−7/2.

In addition, a comparison of the plots shows that the ν_B_^−1^-dispersion of the FDSs is smoothed out as D increases (see the shallowing of 2e_1_ level “depth” at ν_B_^−1^~0.5), which shows a suppression of the quantum confinement effect. 

The data in Figure 4a–c show that the ems are observed in the p- and d-single particle states forming B-em-Cs over the entire B_e_^n^-range measured. The composites correspond to a set of n GSs |ν_nS1_^+^ > ^N^_1/νB_ given in Table 1. In all B-em-Cs, only the 2e s-state is em-free and has ν = 2 (e^2^_0_-state) for n = 1 and 2 and ν = 1 (2e^1^_0_-state) for the n=3 composites. The rest are B-em states, which have ν = 1 and 2/3 for 2e p_y_/p_x_-states and ν = 1/2, 1/3, 14 and 1/8 for 1e 10(p_x_)-, 02(d_y_)-, 20(d_x_)- and 03(f_y_)–states, depending on B, N^*^ and D.

The n = 1 structures in the inserts in Figure 4a–c show that in the D01m dot B-em^M^-C has two single-state Majorana ems (em^M^) (see Section C.1) e^1^_-1_(0) and e^1/4^_−2_(0) and one two-state em^M^, formed by s-e and e^2/3^_1_ states, in which the V of the former is compensated by the aV of the latter. Note, however, that this compensation is not complete since two s-es in the ν = 2 state generate two-times smaller flux. In the D07m dot, the n = 1 em^M^-C has the same number of Vs N_V_ since the extra e generates e^1/2^_2_(0) em^M^ in a d_x_-state taking two Vs from the d_y_-state. In the D09m dot, MMs e^1/2^_−2_(2) and e^1/2^_2_(2) are generated in d_x_- and d_y_-states increasing the N_V_ on four Vs/aVs. 

For *n* = 2 and *n* = 3, *em*^M^-C are formed by the addition of *V*s and elimination of *aV*s, resulting in an increase/decrease of *N_V_*/*ν_n_*_S_^+^. Thus, we observed the expected *N_V_* increase, with an increase of *D* and *B*, and the minimum *N_V_* value of 9 has *n* = 1 *em*^M^-C in D01m, while the maximum of 21 has *n* = 3 *em*-C in D09m.

The charge values obtained are in the range *e_N_ =* (3−6)*e*_0_, which is thus up to two times smaller than that without *em*s, i.e., (7–8)*e*_0_. The reduced charge for *n* = 1 corresponds to a zero field *B_ν_* generated by the composite, i.e., *ν*_1*S*_^+^ = ∞, and thus can be considered as Majorana anyon/*em*-C (*em*^M^-C). This *em*^M^-C has *e_N_^*^* = 4.91, 5.66, and 4.91 in the dots D01m, D07m and D09m, respectively, which shows that the increase of *e_N_* of D07m relative to D01m due to an extra *e* is compensated in D09m by its decrease due to the increase of the *N_V_*. 

### 3.5. Charge Hysteresis and Majorana Modes 

The Δ*_ν_* values calculated show an absence of the steps in the D01m and D07m dots and their presence in the D09m dot (see the corresponding curves in Figure 4a–c). Zero Δ*_ν_* (no steps) occurs when the photo-excited (topmost) *em* states and the initial states of the neighboring *em*-Cs have the same *ν* and *N_V_*, respectively (see Equation (A9)). This is the case for the smallest D01m dot for both *B_e_*↑ and *B_e_*↓ measurements. 

For the intermediate size D07m dot, this took place only in the *B_e_*↑ data, and in the *B_e_*↓ data a negative jump occurs at *ν_B_*^−1^~0.27 (*ν*~7/2). After the jump, the low-energy position of the *s*-peak is maintained under a further field decrease down to zero. This shows that two *em*^M^-Cs exist in the initial state in the D07m dot, which are 3(∞) and 19/7(∞) *em*^M^-Cs (see *V*-structure in the lower right insert in Figure 4b), among which the latter is a MM of the former appearing because of the generation of an *e*^1^_−1_(0) *em*^M^. The *B_e_*↓-jump is thus related to a charge increase in the initial state due to *N_V_* reduction in the 3(∞)-*em*^M^-C and is reproduced well by *E^em^*^-C^_s_(*B*). Thus, *B_e_*↑–*B_e_*↓ measurements reveal charge hysteresis (CH). CH is accompanied by *B_e_*↑–*B_e_*↓ *p*-peak splitting at *ν_B_*^−1^, *B* = 0, which can indicate generation of additional *V*s in the *p*-state. 

In the largest dot D09m the *ν_l_* values of the four topmost *em* states are 1/4_2_, 1/8_−2_, 1/2_1_, and 1/3_−3_, which are different and thus give three Δ*_ν_* steps at *ν* = 7/2, 2 and 5/4, the first and the third of which are negative, as is seen in Figure 4c. The number and position of these steps match well to that of the *B_e_*↑ experiment, but the sign of the first and third jumps is positive instead. This implies a MM in the corresponding initial state, substituting *V*s of the photo-excited *em*. Analysis has shown that this is the *e*^1^_−1_(0) *em*^M^ (the same as in the D07m dot) forming the *e*^2/5^_−1_(1) *em* from the e^2/3^_−1_(1) one. The calculated *E*_s_*^em^*^-C^ accounting for this MM shows good agreement. The *B_e_*↓ data show two CH starting at *ν~*5/4 (CH_1_) and 2 (CH_2_), respectively. In the CH_2_
*em*-C, there is an increase of *p*-peak energy, indicating the generation of additional *V*s.

### 3.6. Collapse of S-Magneto-Electron-Composites

The structure of *S*-*em*-Cs and the *ν*_0_^+^(*B_ν_*) values can be derived from the crossing of the positions of the *p*- and *d*-peaks at *B_e_* = 0 and *em*-FDSs of the *B*_n_-range. Thus, a position of the topmost *d*-peak corresponds to a *d_x_*-state at *ν* ~5/2 and *S* = 3/2 in the D01m dot and to *d_y_*-state at *ν*~5/2 and *S* = 0 in the D07m and D09m dots. Note that while for the D01m dot the total spin of the mirror *aem S-* and *B*-composites is the same (3/2), for the D07m and D09m dots it is different (0 versus 1 and 2). Note also that the *ν*_0_^+^ value of all dots (see Table 1) are nearly the same, i.e., ~5/2 (for dots D01m and D07m *ν*_0_^+^ = 5/2 + 1/8 and 5/2 + 1/6, respectively), and the corresponding *B_ν_* values are larger than *B_ec_* on ~1 T.

The preference of the self-formation of the *S*-*em-*Cs follows from their smaller charge compared to the *B*-*em*-Cs; it is 3.75*e*_0_, 3.05*e*_0_ and 2.08*e*_0_ for the *S*-*em-*Cs versus 3.91*e*_0_, 5.41*e*_0_ and 3.85*e*_0_ for the *B*-*em-*Cs for the D01m, D07m, and D09m dots, respectively. Charge reduction is provided by the extra *V*s in the *p*-states (see the corresponding *V*-structures in the left inserts in Figure 4a–c) and thus there can be some mechanism for selecting the *V* direction in these states. These preferential *V*s create *e*^*1/2^_−1_(2) and *e*^*1/6^_1_(4) *S*-*em*s (we add a superscript * to distinguish *S*-*em*s from *B*-*em*s). 

In the D01m dot, a symmetric positive *s*-jump at *ν_B_*^−1^~0.1 related to *S-em*-C destruction indicates the generation of *aV*s/*V*s decreasing/increasing *B_ν_* needed for formation of *n* = 1 *B-em-*C/*S*-*em*-C in *B_e_*↑/*B_e_*↓ measurements. A transition to *S*-*em*-C for the *B_e_*↓ data is accompanied by a red shift of the *p*-peak, indicating emission of the *p_y_*-component, which is not seen for *B*-*em*-Cs. Its appearance implies a redistribution of the hole density (see (Section A.3) under *V* formation and the instability of *B*-*em*-C. 

In the D07m and D09m dots, negative *B_e_*↑ *s-*jumps correspond to the annihilation of MMs under a decrease of *B* and their absence in the *B_e_*↓ data indicate a suppression of the MM under the collapse of *n* = 1 *B-em*-Cs. The signatures of *S*-*em*-C instability in these dots is a *B_e_*↑–*B_e_*↓ splitting of the *p*-peak, which is stronger in D07m dot.

## 4. Discussion

### 4.1. Majorana Anyons

A subset of the GSs measured is different from that obtained in the exact quantum mechanical CI calculations in Section B.1. For D09m dot, they are |∞_1_, 3_2_, 4/3_2_ > ^8^_1.1_ (see Table 1) and |8_0_, 7/2_1_, 7/3_0_, 2_1_ > ^8^_0.6_ (see Figure A2a,b), respectively, showing a reduction of the number of *B-em*-C GSs compared to *e*-GSs due to a suppression of the level crossing (see Section C.2). The CI set includes fractional *ν* GSs, which, however, do not imply a generation of *V*s by single electrons, i.e., fractional charge, and reflect only a matching of the number of *V*s to the total angular momentum value [36,37]. Moreover, the observed *B*-*em*-Cs have a variety of MM states, including *d*-type MMs, *em*^M^ and *em*^M^-Cs, which is unexpected and very important accounting for the limited experimental data on MMs (mainly for the *p*-type-states) in FQHE [25], one-dimensional hybrid superconductor topological structures [38,39] and in topological superconductor [40]. Thus, our observations give high impact to the physics of anyon/Majorana states and to the perspectives of their use for TQC. Note that from five *em*-Cs observed there is only one MM-free, which is a *ν*_2,1_^+^ = 5/2 composite in the D07m dot, and, thus, MMs seen are intrinsic features of the confined electrons for *ν* > 1, in contrast to *ν* < 1 molecular states [16]. In the context of *em*^M^ and *em^M^*-Cs, a SC analog of such states has been reported in the compound EuFe_2_(As_1−x_P_x_)_2_ [41]. 

The Δ*_ν_* steps observed demonstrate detection of fractional charge variations related to MMs using PL spectra. Moreover, the observed CHs are related to the states with and without MM, i.e., to a MM qubit (see Appendix D), and for the D07m and D09m dots these are zero-field qubits. This two-level qubit can be used in conventional schemes of quantum computing and/or quantized information bit of classical Boolean logics [42]. The data show that the appearance of the Δ*_ν_* steps depends on *D*, *N* and *B*, or more generally on *N_V_* and *r_s_*, and is suppressed with *r_s_* decrease. This gives the opportunity for engineering of MM qubits. The CH is related to the suppression of *aV* generation under a decrease of the field and evidently reflects some fundamental interaction between *em V*s and a sign of the magnetic field variation, which needs further investigation.

### 4.2. Magneto-Electrons and Fractional Quantum Hall Effect States

Our experimental results presented here and in the previous studies [15,16] show that the fractionally charged *em*-anyons emerge in the quantum confined (gapped) *e*-states, which allows superconducting-type sub-*e* PCLs/*V*s generated by a single or a pair of *e*s, creating *em*s. This allows consider a description of the FQHE in terms of a liquid crystal (LC), rather than as an incompressible liquid, in which specific *ν_nS_*^+^-*B-em*-C is a unit cell of the crystal. The nucleation of the *ν*_n*S*_^+^-*B*-*em*-C LC is provided by a quasi-ordering of potential fluctuations (PF) [43] of the corresponding size as was suggested by us in [15]. The experimental evidence of such quasi-ordering is the observation of the Wigner-crystal-type network patterns of the localized electron states for *ν* = 1, observed in a scanning electrometer probe experiment in quantum Hall 2D-*e* structures [44]. 

Within the LC description, the conductivity plateaus are quantum-confined-type *L^S^* plateaus (see Figure A2b,d) rather than a set of states above the mobility edge of the disorder-broadened LLs. The *L^S^* plateaus strongly overlap, as follows from the *E_L_^S^* curves in Figure A2e, which is consistent with the models describing the temperature-dependence of the width of IQHE and FQHE plateaus [45]*,* revealing a Lorentzian broadening of LLs. Moreover, LC description does not involve *e-e* interaction as shown in Appendix B.

We can suggest that the *em*^M^ Majorana anyons revealed are the CFs of the quasi-particle description of the FQHE and they naturally explain the zero internal field of CFs, leading to a termination of the conductance plateaus at even fractional filling factors. The description of CFs using LC implies *B-em*-C consisting of *e*^1/2^_0_ and *e*^1/2^_−1_(0) *em*s, in which the latter contribute to the conductivity. The *S-em*-C of such type can explain the ½ and ¼ fractional quantization of the conductivity of the holes and electrons in Ge [46] and GaAs [47] nanowires, respectively, in a zero magnetic field and the “0.25 anomaly” of the conductivity of a point quantum contact [48,49]. 

The signatures of *em*^M^s in these experiments using confined geometries, i.e., probing of submicron areas, imply that the corresponding PFs are naturally formed in 2D-*e* structures and that their short- versus long-range order arrangement determine the formation the corresponding LCs and FQHE conductivity plateaus. Thus, in the extremely pure, high mobility structures in which FQHE plateaus emerge, the imperfection density is in a dilute regime (~1%), which provides a wide spectrum and long-range order of the PFs [43], induced by *S*- and *B-em*-Cs. The PFs depth in 2D-*e*-structures is ~10 meV [50], which is an order of magnitude smaller than in InP/GaInP_2_ QDs, and we can expect that the FQHE LC states will include *em*-Cs having size ~50 nm, adopting only *s*- and *p*-*em*s, and assume extended imperfections having a size of at least a few nanometers.

### 4.3. Majorana Anyons and Topological Quantum Computing

The proposal of the topologically protected qubit involving the *ν* = 5/2 state [31] is based on Laughlin’s quasi-particle description of FQHE states [14], in which this state is assigned to a non-Abelian topological phase characterized by a Pfaffian wave function [51,52]. This state is considered as a set of degenerate *p*-wave paired fermions having charge *e*_0_/4. The qubit is formed by a zero-energy Majorana excitation of a fused fermion having charge *e*_0_/2. We can suggest that the *em* analog of the *e*_0_/4 fermion is *e*^2^_0_ + *e*^1/8^_−1_(2) *B*-*em*-C, the *e*_0_/2 state is *e*^2^_0_ + *e*^1/8^_−1_(2) + *e*^1/8^_1_(2) *B*-*em*-C, formed by addition of two *e*s, and MM is *e*^2^_0_ + *e*^1/8^_−1_(2) + *e*^1/8^_1_(0) *B*-*em*-C. It follows from our results that it may be possible to realize and measure such *B*-*em*-Cs in InP/GaInP_2_ using appropriate *N* and *D*, and to check the formation of the corresponding LC states in 2D-*e* structures using high-spatial-resolution optical [16] and electrometrical [53] measurements. Moreover, it may be possible to realize similar states using *S*-*em*s, thus creating magnetic-field-free TQCs. 

## 5. Conclusions

We have used the magneto-PL spectra of InP/GaInP_2_ QDs having seven electrons and a Wigner–Seitz radius ~1.3 to demonstrate the self-formation of fractionally charged anion(magneto-electron) composites having *ν*~5/2 and Majorana modes/anyons. We observed the destruction of these composites and the appearance of anti-*em* composites induced by a magnetic field. Using a theoretical description of magneto-PL spectra and an analysis of the electronic structure of these dots based on single-particle Fock–Darwin spectrum and many-particle configuration interaction calculations, we show the formation of fractionally charged anion states based on the mechanism in which non-interacting quantum confined electrons generate sub-electron persistent current loops (vortexes). This implies the liquid-crystal nature of the fractional quantum Hall effect states, in which Majorana anyons are spontaneously formed, and opens new perspectives for the realization of topological quantum computing. 

## Figures and Tables

**Figure 1 nanomaterials-12-01016-f001:**
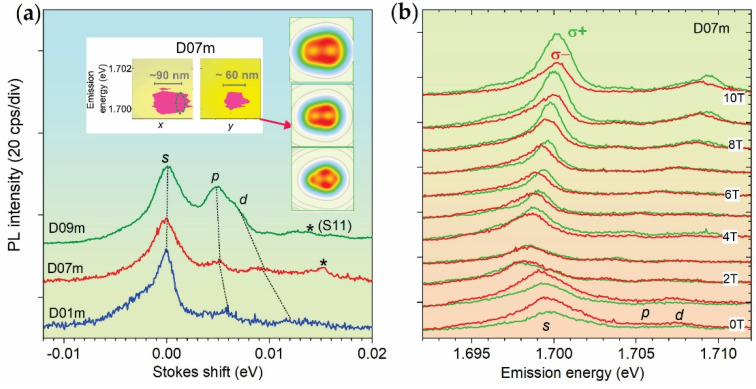
(**a**) Comparison of low-temperature (10 K) PL spectra of D01m, D07m and D09m InP/GaInP_2_ QDs. Lines connect the peak maxima and stars denote neighboring dots. Center inserts show portions of contour plots of a set of spatially resolved near-field PL spectra of D07m measured along two perpendicular directions, in which regions corresponding to the size of emission area are outlined by the magenta color and the horizontal bars. The dashed ellipse in *x*-scan outlines contribution of neighboring dot. Right inserts show charge density distributions (size 100 × 100 nm) of the dots (upper D09m, lower D01m) calculated using the CI method. (**b**) Circular polarized spectra of D07m measured in the range *B_e_* = 0–10 T.

**Figure 2 nanomaterials-12-01016-f002:**
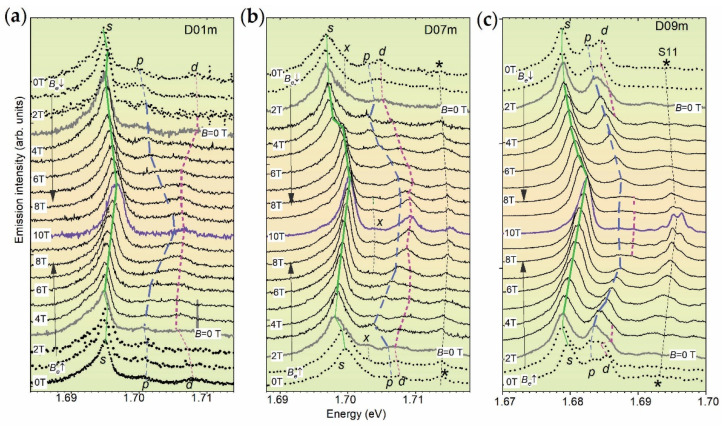
Low-temperature (10 K) magneto-PL spectra (thin solid curves) of D01m (**a**), D07m (**b**) and D09m (**c**) QDs measured in the field range *B_e_* = 0–10 T (thick solid curves are for *B* = 0 T, short-dot curves are for |*B_e_*| < |*B*_ec_|). Solid, dashed and short-dashed lines (thinner for |*B_e_*| < |*B*_ec_|) connecting the maxima of *s*-, *p*- and *d*-peaks, respectively, are drawn to outline their shift and bends; a star symbol denotes neighboring dot, which in (**c**) is the single electron dot S11 studied in Ref. [15]. Vertical arrows at the left show direction of field increase.

**Figure 3 nanomaterials-12-01016-f003:**
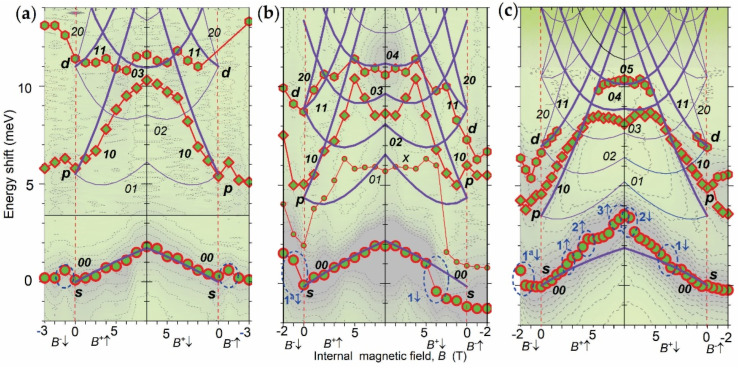
Peak shifts (circles −*s*, diamonds −*p*, hexagons −*d,* small circle −*x*) versus internal magnetic field *B* and calculated shifts (solid curves) overlaid on the PL spectra contour plot of D01m (**a**), D07m (**b**) and D09m (**c**) InP/GaInP_2_ QDs. Changes of *B*–field has four ranges: decrease from *B_ν_* to 0 T (*B^−^*↓), increase from 0 to 10 T-|*B_ν_*| (*B^+^*↑), decrease from 10 T-|*B_ν_*| to 0 T (*B^+^*↓) and increase from 0 T to *B_ν_* (*B^−^*↑). Dashed ovals mark *s*-shift jumps.

**Figure 4 nanomaterials-12-01016-f004:**
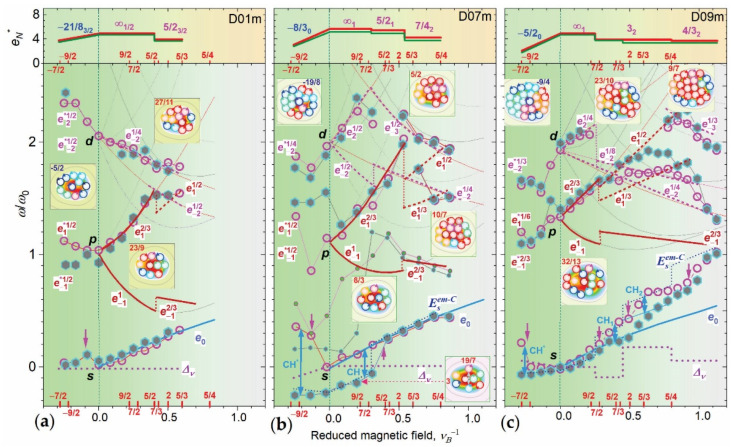
Experimental (empty circles/solid hexagons are for *B_e_*↑/*B_e_*↓ measurements) and fitted (thick solid/dashed curves are paired/single *e*-states, respectively) *s*-shifts and *e^ν^_l_*-FDSs (lower plots) and *e_N_*^*^ (upper plots) versus *ν_B_*^−1^ for D01m (**a**), D07m (**b**) and D09m (**c**) InP/GaInP_2_ QDs. Insets are *V*-structures of the *em*-Cs represented by *N_V_ V*s arrangement (arrowed circles) overlaid on the dot CDD. Numbers on *ν_B_*^−1^ axis, total charge curves and inside insert’s frames are *ν*, *ν_nS_*^+^ and *ν_nS_* values, respectively. Arrows (one/double sided), thick and thin dotted curves on *s*-shifts are psotion/hysteresis of the jumps, Δ*_ν_* and *E_s_^em^*^-C^ functions, respectively (see text). *V*s circles color code is for *B*-*em*s– dark red, yellow/violet and red are *V*s in *s-*, *p*- and *d*-states and light blue/green and blue are *aV*s in *p*- and *d*-states; for *S*-*em*s the colors for *V*s and *aV*s are interchanged. The insert at the lower right corner of (**b**) is the *V*-structure of *n* = 1 in initial state in B*_e_*↓-range in which dashed circles, marked by arrows, correspond 19/7 state (see text).

**Table 1 nanomaterials-12-01016-t001:** Properties of anyon(magneto-electron) composites of InP/GaInP_2_ quantum Hall puddles.

QD	*N**	*ħ**ω*_0_^*^/*ħ**ω*_0_ (meV)	*D* (nm)	*r* _s_	*B*_ec_/*B**_ν_* (T)	*ν*_0_^+^(*ν*_0_)*_S_^eN*^*	*ν_n_*^+^(*ν_n_*)*_S_^eN*^*		
D01m	7	5.5/9.2	65	1.2	3/3.6	−^5^/_2_(−^21^/_8_)_3/2_^3.75^	^23^/_9_(∞)_1/2_^4.91^	^27^/_11_(^5^/_2_)_3/2_^3.91^	-
D07m	8	4.5/7.5	75	1.3	2.5/3.2	−^21^/_8_(−^8^/_3_)_0_^3.05^	^8^/_3_(∞)_1_^5.66^	^5^/_2_(^5^/_2_)_1_^5.41^	^10^/_7_(^7^/_4_)_2_^4.25^
D09m	8	3.5/5.8	85	1.5	2/3	−^9^/_4_(−^5^/_2_)_0_^2.08^	^32^/_13_(∞)_1_^4.91^	^23^/_10_(3)_2_^3.88^	^9^/_7_(^4^/_3_)_2_^3.71^

## Data Availability

All data needed to evaluate the conclusions in the paper are present in the paper.

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
