# Peer review of "Majorana Anyon Composites in Magneto-Photoluminescence Spectra of Natural Quantum Hall Puddles"

_nanomaterials, 2022, doi:10.3390/nano12061016_

Round 1
Reviewer 1 Report
In the presented work entitled „Majorana anyon composites in magneto-photoluminescence spectra of natural quantum Hall puddles” by A. M. Mintairov et al., the Authors conduct analysis of the spectra of quasi-two dimensional quantum dots and reveal suppression of magnetic field dispersion, paramagnetic shifts, jumps of the energy of the emission components for filling factors as well as observe the magnetic hysteresis of the jumps and a dependence of all these anomalous features on the Wigner-Sietz radius. As a result, the Authors claim that these observations can be described by the raidus-dependent formation of the anyon composites and that the anyon states observed involve Majorana modes (MMs), which can be considered as Majorana anyons.
In general, the article presents results and discussion that may be of interest to the scientific community interested in realization of the topological quantum computing. In other words, the presented paper considers timely subject. From the technical point of view, the paper is well written and the analysis seems to well-constructed and free of errors. Still, I would like the Authors to address the following points:
- The core aspect of the presented manuscript is that the Authors consider quasi two-dimensional quantum dots (2 QDs). In particular they deal with the InP/GaInP2 However, is there any chance to observe at least some of the reported phenomena in other types of the 2D QDs? What are the requirements to be able to observe Majorana anyons in 2D QDs made of other materials? For example, can the Authors briefly consider semiconducting materials such as transition metal dichalcogenides that can be used to prepare 2D QDs (see J. Mater. Chem. B 6, 8011-8036 (2018)) and presents rich magnetic behavior (see e.g. valleytronic properties in Phys. Rev. B 101, 115423 (2020))? I believe that such discussion would be instructive for the readers and would improve context of the presented results.
- In section 2.2 the Authors refer to Refs. 16, 33, 34, in terms of the spectroscopy measurements. I believe that it is a bad practice to refer readers to other papers in order to present them with the description of the employed methods. At least brief description of what can be found in the aforementioned references should be presented in the presented manuscript. Moreover, in section 2.3, I believe that mentioning OriginLab software requires citation (at lease hyperlink to the software website).
- The Authors should decide what kind of formatting use when referring to the Figures, especially subfigures, i.e. the sub-figures are marked using brackets (e.g. 1 (a)) whereas in the text the Authors are not using brackets when referring to the subfigures (e.g. 1a). Please correct.
- There are at least seven references by A.M. Mintarov, and I believe that some of these auto-citations may be reduced e.g. 33 and 34 by describing more in details employed methods in the present paper (see comment 2). I urge the Authors to reduce auto-citations where necessary and try to use more diverse sources.
- There are places in the text where it seems like the Authors used multiple spaces. Please correct.
*end of report*
Author Response
- The referee’s question includes two points: “What are the requirements to be able to observe Majorana anyons in 2D QDs made of other materials … and can the Authors briefly consider … transition metal dichalcogenides …?”
The only “requirements” are the value of the Wigner-Seitz radius rs, which should be ~1.5, and the number of electrons, which should be >2. For a specific material system, rs is determined by the size and the number of electrons in the QD (see columns 2, 4 and 5 of Table 1 for our material system), which is determined in turn by the Bohr radius of the material. Thus, Majorana anyons 2D QDs can exist in any material system, including transition metal dichalcogenides, provided the requirements are fulfilled. It follows from Section 4.2 of our paper that the direct evidence of the implementation of these requirements is the observation of the fractional quantum Hall effect. In section 4.2 we discuss the signatures of Majorana anyon QDs, formed in III-V and II-VI materials beyond the InP/GaInP system, observed in confined quantum Hall structures in conductivity experiments (Refs [47-50]).
A recent observation of the fractional quantum Hall effect in WSe2 [Nat. Nanotechnology 15, 569 (2020)] guarantees the possibility of realizing Majorana anyons QDs in transition metal dichalcogenides. However, the problem with these materials in general, and with colloidal QDs (mentioned by Referee) in particular, is a high density of intrinsic defects, which effectively reduces rs and prevents doping. These defects seem to give the main contribution in Coulomb blockade experiments in gated QDs reported so far [see for example Phys. Rev. Appl. 13, 054058 (2020)], preventing the formation/observation of Majorna anyons in these materials.
While we appreciate this interesting and important question, we think it is not directly relevant to our results, since most of the issues under the question are presented in the paper and since a discussion of the transition metal dichalcogenides materials is outside the scope of the paper. We also should point out that we are working on the realization of Majorana anyon QDs in these materials and we have developed methods of locally doping them, which will be published elsewhere and in which the questions raised by Referee will be discussed for broader audience.
- We agree with this point and we have added the description of the essential features of the measurements and a hyperlink to the software website.
- We think the point of labeling the subfigures on the Figure/captions and in the main text is not relevant, since there is not a clear requirement of such labeling in the Nanomaterials manuscript template that we used.
- We partially agree with this comment and have removed one self-citation [Ref.35].
- We checked multiple spaces and corrected them.
Reviewer 2 Report
In this work, the authors reported a suppression of magnetic field dispersion, paramagnetic shifts and jumps of the energy of the emission components for filling factors and demonstrated the liquid-crystal nature of the fractional quantum Hall effect states. I believe such data can be helpful for the realization of topological quantum computing area. Therefore, I believe this manuscript contains enough scientific information and can be accepted for publication in Nanomaterials after a careful revision. A few other comments are listed as below:
- In Figure. 1(a), the authors provided charge density distributions (size 100x100 nm) of the dots. But the authors should mark clearly and provide detailed description for the corresponding samples of the included images.
- In the description of Figure 1, the authors declared as “The peaks have a full-width-at-half-maximum ~3 meV”. However, three of those peaks (s, p and d) even cannot be distinguished because of their weak intensity. Please provide the basis of this data and analyze them carefully.
- With the increasing size, the main s peak show no obvious shift, while the peaks of p and d shift significantly. The authors should provide a reasonable explanation for that.
- In Figure.2, s-peaks of D01m, D07m and D09m show significant different peak position, but the position of s-peak remains unchanged in Figure. 1(a). Why?
Author Response
- We have added the marking of CDD of dots D09m and D01m in Figure 1a in Figure caption. The CDD of D07m is already marked.
- The basis of the analysis and a careful description of the spectral parameters is presented in the first sentence of Section 2.3. Specifically, the full-width-at-half-maximum parameter and its precision, mentioned by the Referee, does not affect any conclusions made (see Fig.2). Moreover, we think that a detailed description of this parameter is redundant and will make it difficult to follow/analyze the main results.
- As stated in the first sentence of section 3.1, the energy scale in Figure 1a is a Stokes shift, which is the energy of the spectral features relative to the s-peak. Thus, the s-peak has the value zero for all dots.
- In Figure 2, the energy scale is the emission energy (see point 3).
Round 2
Reviewer 2 Report
This manuscript can be accept in present form